# *VdPT1* Encoding a Neutral Trehalase of *Verticillium dahliae* Is Required for Growth and Virulence of the Pathogen

**DOI:** 10.3390/ijms25010294

**Published:** 2023-12-25

**Authors:** Lihua Chen, Xiaohu Ma, Tiange Sun, Qian-Hao Zhu, Hongjie Feng, Yongtai Li, Feng Liu, Xinyu Zhang, Jie Sun, Yanjun Li

**Affiliations:** 1The Key Laboratory of Oasis Eco-Agriculture, Agriculture College, Shihezi University, Shihezi 832000, China; chenlihua@stu.shzu.edu.cn (L.C.); xhma@stu.shzu.edu.cn (X.M.); suntiange@stu.shzu.edu.cn (T.S.); liyongtai@stu.shzu.edu.cn (Y.L.); liufeng@shzu.edu.cn (F.L.); zhxy@shzu.edu.cn (X.Z.); 2CSIRO Agriculture and Food, GPO Box 1700, Canberra 2601, Australia; qianhao.zhu@csiro.au; 3State Key Laboratory of Cotton Biology, Institute of Cotton Research of Chinese Academy of Agricultural Sciences, Anyang 455000, China; fenghongjie@caas.cn

**Keywords:** cotton, RNA-sequencing, RNA-seq, trehalase, *Verticillium dahliae*

## Abstract

*Verticillum dahliae* is a soil-borne phytopathogenic fungus causing destructive Verticillium wilt disease. We previously found a trehalase-encoding gene (*VdPT1*) in *V. dahliae* being significantly up-regulated after sensing root exudates from a susceptible cotton variety. In this study, we characterized the function of *VdPT1* in the growth and virulence of *V. dahliae* using its deletion-mutant strains. The *VdPT1* deletion mutants (*ΔVdPT1*) displayed slow colony expansion and mycelial growth, reduced conidial production and germination rate, and decreased mycelial penetration ability and virulence on cotton, but exhibited enhanced stress resistance, suggesting that *VdPT1* is involved in the growth, pathogenesis, and stress resistance of *V. dahliae*. Host-induced silencing of *VdPT1* in cotton reduced fungal biomass and enhanced cotton resistance against *V. dahliae*. Comparative transcriptome analysis between wild-type and mutant identified 1480 up-regulated and 1650 down-regulated genes in the *ΔVdPT1* strain. Several down-regulated genes encode plant cell wall-degrading enzymes required for full virulence of *V. dahliae* to cotton, and down-regulated genes related to carbon metabolism, DNA replication, and amino acid biosynthesis seemed to be responsible for the decreased growth of the *ΔVdPT1* strain. In contrast, up-regulation of several genes related to glycerophospholipid metabolism in the *ΔVdPT1* strain enhanced the stress resistance of the mutated strain.

## 1. Introduction

*Verticillium dahliae* is a soil-borne phytopathogenic fungus that causes Verticillium wilt disease in a broad range of plant hosts, including many economically important crops, such as cotton, resulting in reduced yield and quality and huge economic losses [1]. *V. dahliae* can persist in soil for many years as long-living resting microsclerotia, making it difficult to control. Under appropriate temperature and humidity conditions, the microsclerotia can be stimulated to germinate after sensing root exudates from host plants, to form infectious mycelium to enter the roots, and begin to grow and spread in the xylem vessels, causing wilting and yellowing of the leaves of the diseased plants, and eventually causing plant death [2]. With the completion of genome sequencing for *V. dahliae*, several genes associated with fungal growth, development and virulence have been identified in recent years [3,4,5,6]. However, a large number of pathogenicity-related genes are waiting to be further identified given the complexity of the molecular mechanisms of pathogenicity.

Trehalose is a non-reducing disaccharide consisting of two glucose residues linked by an α,α-1,1-glycosidic bond, which has stable physicochemical properties and is widely found in many organisms, including plants, algae, fungi, bacteria, insects, and some invertebrates [7,8,9]. Endogenously, trehalose produced by fungi could be used as stabilizers and protectors of proteins and biological membranes under stress conditions, such as cold, heat, salinity, and ROS stress, in addition to being a reserve material for carbon and energy [10,11,12,13,14,15]. There are four major trehalose catabolic pathways in different organisms, and the dominant catabolic system in fungi is the hydrolysis of trehalose to glucose by trehalase; this pathway has a high substrate specificity for trehalose and hydrolyzes one molecule of trehalose into two molecules of glucose [16,17,18].

According to the Carbohydrate-Active Enzyme Database (CAZy, http://www.cazy.org/ (accessed on 23 October 2022)), trehalases (EC 3.2.1.28) mainly belong to the GH15, GH37, and GH65 families of glycoside hydrolases, but most of the trehalases found so far belong to the GH37 family [19]. Trehalases in fungi are classified as acidic (non-regulatory) and neutral (regulatory) trehalose hydrolases according to their optimal pH and regulatory properties. Acid trehalases mostly belong to the GH65 family and are extracellular enzymes that promote the cellular uptake of exogenous trehalose; they have multiple N-glycosylation sites, N-transmembrane structural domains, or signal peptides [18,20]. Neutral trehalases, mostly belonging to the GH37 family, are intracellular enzymes that generally have a cAMP-dependent protein phosphorylation site, EF-like Ca^2+^ binding domain, trehalase labeling domain 1, trehalase labeling domain 2, and a glycosylphosphatidylinositol membrane-anchoring domain [21,22]. Several genes encoding trehalase have been identified in various fungi, including *Saccharomyces cerevisiae*, *Aspergillus niger,* and *Mycobacterium smegmatis* [10,23,24]. Some trehalase-encoding genes have been found to be associated with fungal growth, virulence, and resistance to stress [23,25,26,27]. The changed hydrolysis of trehalose leads to abnormal physiological activity in organisms [28]; therefore, trehalase is being developed as a new fungicide and insecticide [29,30]. Validamycin A (Val. A) is a trehalase inhibitor and has been used to treat plant diseases caused by fungi [30,31,32,33,34]. So far, no study on the identification and functional characterization of trehalase-encoding genes has yet been reported in *V. dahliae*.

In a previous study, we found a trehalase-encoding gene *VdPT1* (VDAG_03038) that was significantly induced by root exudates from cotton varieties susceptible to *V. dahliae*, but not by root exudates from resistant varieties, implying an important role of *VdPT1* in the pathogenesis process of *V. dahliae* [35]. To gain insight into the role of *VdPT1* in the growth and virulence of *V. dahliae*, in this study, we generated mutated *V. dahliae* strains by deleting the *VdPT1* gene and we investigated the effects of the mutation in *VdPT1* on the growth and pathogenicity of *V. dahliae*. This research will help to deepen our understanding of the pathogenic molecular mechanisms of *V. dahliae*.

## 2. Results

### 2.1. Identification of Trehalase-Encoding Genes in V. dahliae

In our previous study, some *V. dahliae* genes were found to be up-regulated after *V. dahliae* sensed root exudates from a cotton variety susceptible to the pathogen [35]. One of those genes was VDAG_03038 or *VdPT1*, which encodes a periplasmic trehalase. *VdPT1* was predicted to contain a 2049 bp open reading frame encoding a trehalase with 682 amino acids (with a trehalase domain (PF01204) at 50-627aa) (Figure 1A). In this study, five additional trehalase-encoding genes (VDAG_04410, VDAG_08616, VDAG_00408, VDAG_01953 and VDAG_03528) were identified in the *V. dahliae* (strain *VdLs.17*) genome. The protein sequences of all six *V. dahliae* trehalases, together with 58 trehalases from other microorganisms, including fungi and bacteria, were used to generate a phylogenetic tree (Figure 1B) in which the 64 trehalases were grouped into three subfamilies (GH15, GH37 and GH65), and *VdPT1* was included in the GH37 subfamily. It was reported that the GH37 subfamily members are mostly neutral trehalases, whereas the GH65 subfamily members are mostly acidic trehalases [19]; therefore, VdPT1 was tentatively presumed to be a neutral trehalase. Like other neutral trehalases of the GH37 subfamily proteins, VdPT1 contains a trehalase labeling domain 1 (PGGRFXEXYXWDXY) and a trehalase labeling domain 2 (QWDXPX[G/A]W[P/A/S]P) (Figure 1C), which were not found in the GH65 subfamily members [22].

### 2.2. VdPT1 Deletion Mutants Had a Reduced Trehalase Activity and Increased Trehalose Content

To investigate the role of *VdPT1* in the growth and pathogenicity of *V. dahliae*, we generated two *VdPT1* deletion mutants (*ΔVdPT1*-1 and *ΔVdPT1*-2) by using a homologous recombination method. The expected deletion in the two mutants was confirmed by PCR (Appendix A) and the expression level of *VdPT1* in the two mutants was significantly lower than that of wild-type Vd911 (Figure 2A). We also generated a strain (*ΔVdPT1-C*) in the background of *ΔVdPT1* in which *ΔVdPT1* was complemented by a functional copy of *VdPT1* (Appendix A). The expression level of *VdPT1* in *ΔVdPT1-C* was restored to normal (Figure 2A).

VdPT1 is a neutral trehalase, presumed to be responsible for the hydrolysis of endogenous trehalose. We compared the trehalase activity and trehalose content in the wild-type (WT) strain (Vd991) with those in the *ΔVdPT1-1* and *ΔVdPT1*-*2* mutants and found that the WT trehalase activity and trehalose content were significantly lower and higher than in the mutants, respectively, and that both trehalase activity and trehalose content of the complementary strain *ΔVdPT1-C* were similar to those of WT (Figure 2B,C), confirming *VdPT1* as a trehalase responsible for catalyzing trehalose into glucose in *V. dahliae*.

### 2.3. VdPT1 Deletion Mutants Had a Reduced Colony Growth Rate and Increased Melanin Production

The wild-type Vd991, its two deletion mutants (*ΔVdPT1-1* and *ΔVdPT1-2*) and the complementary strain (*ΔVdPT1-C*) were grown on PDA, CM, or BMM medium to compare their growth and pigmentation. Although no differences in the colony morphology, colony growth rate, or melanin formation were observed between WT and *ΔVdPT1-C*, the two deletion mutants were darker-colored and had a higher melanin production on PDA and BMM media (Figure 3A). After 21 days of incubation on the PDA or CM medium, the colony diameter of the two deletion mutants was reduced significantly compared with that of WT (by 21–22.1% on CM medium and by 5.2–7.4% on PDA medium), but a reduction was not observed when the strains were grown on BMM medium (Figure 3B).

### 2.4. VdPT1 Deletion Led to a Delay in Mycelial Growth and Impaired Mycelial Penetration

We further compared mycelial growth of the four *V. dahliae* strains after 12, 24 and 48 h of incubation on PDA medium. Compared to WT and *ΔVdPT1-C*, the two deletion mutants produced significantly fewer mycelium after 12 and 24 h of incubation, although their amount of mycelium caught up after 48 h of incubation (Figure 4A). In addition, the mycelial-penetration ability of the two mutant strains seemed to be weaker than that of WT and *ΔVdPT1-C* (Figure 4B), because the colony size of the two mutants was significantly smaller than that of WT and *ΔVdPT1-C* at 7 days following removal of the cellophane on which conidia of each strain had grown for 7 d. These results suggested that *VdPT1* is required for mycelial formation and penetration of *V. dahliae.*

### 2.5. VdPT1 Deletion Resulted in a Delay of Conidial Germination and Reduced Conidial Production

Monoconidial growth of the four strains was observed after 12, 24, and 36 h of incubation on PDA medium. It was found that monoconidial germination of the two deletion mutants (*ΔVdPT1-1* and *ΔVdPT1-2*) was slower than that of WT and *ΔVdPT1-C*. After 12 h of incubation on PDA medium, significant monoconidial germination was found in WT and *ΔVdPT1-C* but not in the two deletion mutants (Figure 5A). After 24 and 36 h of incubation on PDA medium, conidia of WT and *ΔVdPT1-C* produced significantly more mycelium than those of the two deletion mutants (Figure 5A). After 12 h of incubation in CM liquid medium, the conidial germination rate of the two mutant strains (61.5% and 57.6%) was much lower than that of WT (82.5%) and *ΔVdPT1-C* (80.6%) (Figure 5B). The conidial production of the two deletion mutants was significantly lower than that of WT and *ΔVdPT1-C* after 24, 48, and 72 h incubation in CM liquid medium (Figure 5C). Collectively, the results indicated that *VdPT1* is involved in conidial germination and production.

### 2.6. VdPT1 Deletion Increased Stress Resistance of V. dahliae

Trehalose helps organisms combat extreme environments, including high temperature, cold, dehydration, and hypertonicity, among others [19]. We tested the responses of the four *V. dahliae* strains to various stresses by growing them under high- or low-temperature conditions, or by supplementing the PDA medium with NaCl, KCl, sorbitol, or validamycin A (Val. A). The deletion mutants (*ΔVdPT1*-*1* and *ΔVdPT1*-*2*) grew better (with larger colony diameter) than WT and *ΔVdPT1-C* under the stress conditions of high temperature, NaCl, sorbitol, and Val. A (Figure 6A,B), although KCl and LT seemed to have no significant impact on the growth of the four strains, suggesting that *VdPT1* is involved in the stress responses of *V. dahliae*.

### 2.7. VdPT1 Deletion Reduced the Pathogenicity of V. dahliae

To determine the role of *VdPT1* in the virulence of *V. dahliae*, we infected the upland cotton variety Xinluzao 7 that is susceptible to *V. dahliae* with each of the four strains. All infected plants showed wilting symptoms at 18 dpi (days post infection) and 30 dpi, with milder symptoms observed in the plants infected with either of the two deletion mutants (*ΔVdPT1-1* and *ΔVdPT1-2*) (Figure 7A). The disease index (DI) of cotton plants infected with the two deletion mutants (*ΔVdPT1*-*1* and *ΔVdPT1*-*2*) was significantly lower than that of the cotton plants infected with wild-type or *ΔVdPT1-C* at 18 dpi and 30 dpi (Figure 7B). In line with this observation, the cotton plants infected with the two deletion mutants had a significantly lower fungal biomass, compared with the WT and *ΔVdPT1-C* at 21 dpi (Figure 7C). These results indicated that *VdPT1* is required for the pathogenicity of *V. dahliae.*

### 2.8. Host-Induced Silencing of VdPT1 Increased the Resistance of Cotton to V. dahliae

TRV-based host-induced gene silencing (HIGS) was adopted to silence the *VdPT1* gene in *V. dahliae*. When the two cotyledons of cotton seedlings were fully expanded, the *Agrobacterium tumefaciens* solution containing pTRV2-*00*, pTRV2-*VdPT1* or pTRV2-*GhCHLI* was injected into cotyledons. When the true leaves of cotton seedlings treated with pTRV2-*GhCHLI* showed bleaching (Appendix A), Vd991 conidial suspension was used to infect TRV-treated cotton by the root irrigation method. At 14 dpi, all infected plants showed wilting symptoms, but the leaf yellowing and wilt symptoms and vascular browning were milder in the plants treated with pTRV2-*VdPT1* compared to the plants treated with pTRV2-*00* (Figure 8A,B). The disease index (DI) of the cotton plants treated with pTRV2-*VdPT1* was significantly lower than that of the cotton plants treated with pTRV2-*00* at 14 dpi and 21 dpi (Figure 8C). Consistent with the phenotypic observations, the expression level of *VdPT1* in different tissues of seedlings treated with pTRV2-*VdPT1* at 21 dpi was significantly lower compared with those treated with pTRV2-*00* (Figure 8D), and the fungal biomass in leaves from 21 dpi plants was significantly lower in the pTRV2-*VdPT1* treated plants than in the pTRV2-*00* treated plants (Figure 8E). These results suggested that *VdPT1* is involved in the pathogenic process of *V. dahliae*, and HIGS of *VdPT1* alleviates the pathogenesis of *V. dahliae* and enhances the resistance of cotton to the pathogen.

### 2.9. RNA-Seq Analysis

To elucidate the physiological role of *VdPT1* in *V. dahliae*, transcriptomic sequencing was conducted to compare gene expression between the WT and *ΔVdPT1*-*1* strains (PRJNA1007260). Both fungal strains were cultured in 200 mL CM liquid medium for 4 days and collected for isolation of RNA to be used in RNA-seq analysis. A total of six RNA-seq libraries (Vd991-1, Vd991-2, Vd991-3, *ΔVdPT1-1*a, *ΔVdPT1-1*b and *ΔVdPT1-1*c) were generated and used in the identification of differentially expressed genes (DEGs). The reliability of the RNA-seq data was confirmed by qRT-PCR analysis of ten randomly selected genes (Appendix A).

A total of 3130 genes differentially expressed between wild-type Vd991 and *ΔVdPT1*-*1*, including 1480 up-regulated and 1650 down-regulated genes in the *ΔVdPT1*-*1* mutant, were identified. To characterize the potential functions of DEGs, we performed gene ontology (GO) enrichment analysis separately for the up-regulated and down-regulated DEGs. The main molecular function GO terms of the down-regulated DEGs were ‘hydrolase activity, hydrolyzing O-glycosyl compounds’, ‘NADP binding’, and ‘DNA binding’. The biological process GO terms of the down-regulated DEGs were chiefly related to ‘carbohydrate metabolic process’, ‘DNA replication’, ‘DNA replication initiation’, ‘lagging strand elongation’, ‘DNA repair’, and ‘chromosome segregation’ (Figure 9B). The major cellular component GO term of the up-regulated DEGs was ‘integral component of membrane’. In addition, it was notable that DEGs related to ‘lipid biosynthetic process’, ‘lipid catabolic process’, ‘response to chemical,’ and ‘cellular response to chemical stimulus’ were up-regulated (Figure 9A). KEGG cluster analysis revealed that the down-regulated pathways mainly included ‘DNA replication’, ‘mismatch repair’, ‘propanoate metabolism’, ‘base excision repair’, ‘cell cycle’, ‘meiosis’, ‘carbon metabolism’, and ‘biosynthesis of amino acids’ (Figure 9D), while the up-regulated pathways included ‘SNARE interactions in vesicular transport’, ‘glycerophospholipid metabolism’ and ‘steroid biosynthesis’ (Figure 9C).

### 2.10. DEGs Related to Hydrolase Activity Hydrolyzing O-glycosyl Compounds

A total of 28 DEGs related to hydrolase activity that hydrolyzes O-glycosyl compounds were found to be down-regulated in *ΔVdPT1-1*, including many genes encoding cell wall-degrading enzymes (CWDEs), such as exoglucanase (EGY21374, EGY19182); endoglucanase (EGY16447) and beta-glucosidase (EGY20159, EGY15425, EGY16975) involved in cellulose degradation; and alpha-N-arabinofuranosidase (EGY22269), beta-mannosidase (EGY14399), xylosidase/arabinosidase (EGY22190) and beta-xylosidase (EGY17487) involved in hemicellulose degradation. It was notable that the expression of several of these genes was remarkably down-regulated in *ΔVdPT1-1*. For instance, the expression level of an exoglucanase-encoding gene (EGY21374) decreased by 1172-fold, two beta-glucosidase genes (EGY20159 and EGY15425) decreased by 444-fold and 45-fold, respectively, and the alpha-N-arabinofuranosidase gene (EGY22269) decreased by 108-fold (Figure 10). To successfully infect and colonize plant tissues, pathogenic fungi secrete a diverse range of CAZymes, especially CWDEs which play important roles in the penetration, invasion, and pathogenesis of pathogens [36]. Therefore, the significant down-regulation of genes encoding CWDEs may be one of the important reasons for the reduced mycelial penetration and pathogenicity of the *VdPT1-1* deletion mutants.

### 2.11. DEGs Related to Carbon Metabolism, DNA, and Amino Acids Biosynthesis

A total of 41 DEGs related to carbon metabolism were found to be down-regulated in *ΔVdPT1-1*, including several genes involved in glycolysis, the pentose phosphate pathway, and the glyoxylic acid cycle. Glycolysis-related genes included pyruvate kinase (EGY17524), hexokinase (EGY22649, EGY16853), glyceraldehyde-3-phosphate (EGY18756), and enolase (EGY21589) genes, whose expression levels decreased by 4.5–6.3 fold in *ΔVdPT1-1*. Genes involved in the pentose phosphate pathway included transaldolase (EGY19531, EGY21104 and EGY23089), fumarylacetoacetate hydrolase domain-containing protein 2A (EGY23078), ribose-5-phosphate isomerase B (EGY18996), dihydroxyacetone kinase (EGY18997), triosephosphate isomerase (EGY18998), and dihydroxyacetone synthase (EGY14084) genes. Genes associated with the glyoxylic acid cycle included mitochondrial 2-methylisocitrate lyase (EGY21244); 2-methylcitrate synthase (EGY21245) and NAD-dependent malic enzyme (EGY15422 and EGY15423);0 and NADP-dependent leukotriene B4 12-hydroxydehydrogenase (EGY23661) genes (Figure 11A). Additionally, multiple genes related to DNA replication and amino acid biosynthesis were also found to be down-regulated in *ΔVdPT1-1* (Appendix A), likely attributed to the down-regulation of genes related to the pentose phosphate pathway, which provides 5-P ribose for nucleic acid synthesis.

Trehalase hydrolyzes one molecule of trehalose into two molecules of glucose. Decreased trehalase activity in the *VdPT1* deletion mutants increased the trehalose content (Figure 2B). Accordingly, the glucose content in the two deletion mutants (*ΔVdPT1*-*1* and *ΔVdPT1*-*2*) was significantly lower than that in the wild-type Vd991 (Figure 11B).

### 2.12. DEGs Related to Glycerophospholipid Metabolism and Steroid Biosynthesis

A total of 21 DEGs related to glycerophospholipid metabolism were found to be up-regulated in *ΔVdPT1-1*, including genes encoding choline kinase (EGY19710), choline/ethanolamine phosphotransferase (EGY21037 and EGY17370), lipase (EGY22128 and EGY15596), phospholipase D2 (EGY13821), and phosphosterase (EGY20208) (Figure 12A,C). Additionally, a total of 12 DEGs related to steroid biosynthesis were also found to be up-regulated in *ΔVdPT1-1*, and they were mainly related to the ergosterol biosynthesis pathway, including genes encoding sterol O-acyltransferase (EGY23203), C-4 methylsterol oxidase (EGY18771), lanosterol synthase (EGY19026), squalene monooxygenase (EGY19918), and C-5 sterol desaturases (EGY13566 and EGY14824) (Figure 12B,D). Glycerophospholipids and steroids are the main lipids of the bilayer membrane and they play key roles in the response to environmental stress [37,38,39]. Up-regulation of genes involved in glycerophospholipid metabolism and steroid biosynthesis may be associated with the enhanced stress resistance of the *ΔVdPT1-1* strain.

Based on the RNA-seq results, we speculated that the deletion mutation in *VdPT1* altered the composition of glycerophospolipids of *V. dahliae*. To verify this hypothesis, we compared the lipidome between the WT and *ΔVdPT1-1* strains using mass spectrometry of methanol-extracted cells. The content of glycerophospholipids, including phosphatidic acid (PA), phosphatidylglycerol (PG), phosphocholine (PC), phosphatidylserine (PS), phosphatidylethanolamine (PE), and phosphatidyl-inositol (PI) was significantly increased in *ΔVdPT1-1* (Figure 13A). Glycerophospholipids play an important role in maintaining the stability of cell membranes [40,41,42]. Therefore, the increased glycerophospolipid content in *ΔVdPT1-1* might enhance membrane integrity under stress conditions, leading to increased resistance to stress, which was verified by propidium iodide (PI) uptake assay. Compared to the WT strain, the *ΔVdPT1-1* strain showed a 20.6% and 56.3% reduction in cell death (PI-stained cells) following the treatment with zero and 0.7 M NaCl, respectively (Figure 13B,C).

## 3. Discussion

### 3.1. VdPT1 Is a Neutral Trehalase and Is Required for Growth and Pathogenicity of V. dahliae

Trehalase enzymes are classified as acidic and neutral trehalases. Acid trehalases are extracellular enzymes that promote the cellular uptake of exogenous trehalose, whereas the neutral trehalases are intracellular enzymes that hydrolyze endogenous trehalose. In this study, six genes encoding trehalase were identified in *V. dahliae* and clustered into three groups, including GH15, GH37, and GH65. VdPT1, containing the two typical trehalase labeling domains of neutral trehalases, was clustered together with the neutral trehalases from other microoganisms into the GH37 subfamily [19]. Substantial levels of trehalose accumulate in fungi to serve as a storage carbohydrate [43,44]. Trehalose is also an important component of fungal conidia and the main source of glucose in the early stages of conidial germination [43,45]. The synthesis and hydrolysis of trehalose are closely linked to the growth and pathogenicity of fungal pathogens. Trehalose synthesis is important for primary plant infection by *Magnaporthe grisea*, whereas trehalose degradation is required for efficient development of fungi in plant tissue following initial infection [25]. The *TPS1* (encoding a trehalose-6-phosphate synthase) deletion mutant of *Magnaporthe grisea* failed to synthesize trehalose and showed poor sporulation ability and significantly reduced pathogenicity of the pathogen [25]. Neutral trehalase (Tre1)-deficient mutants of *Botrytis cinerea* are unable to mobilize trehalose during conidial germination, thereby reducing conidia germination rates [27]. Deletion of the two trehalases (NTH1 and NTH2) in *C. neoformans* resulted in severe defects in spore production, a decrease in conidial germination, and an increase in the production of alternative developmental structures [26]. In this study, we found that *VdPT1* deletion resulted in decreased colony expansion, delayed mycelial growth and conidial germination, reduced mycelial penetration ability, reduced conidial production and reduced pathogenicity, suggesting that *VdPT1* is required for the growth and pathogenicity of *V. dahliae.*

### 3.2. Down-Regulation of Cell Wall-Degrading Enzyme Genes Is Responsible for the Decreased Pathogenicity of V. dahliae

Plant fungal pathogens need to produce a range of hydrolytic enzymes to facilitate infection and colonization [36,46]. Enzymes capable of degrading the cell wall are often referred to as ‘cell wall degrading enzymes’ (CWDEs). It was reported that the *V. dahliae* genome encodes a large number of CWDEs, mainly pectinases and cellulases to adapt the components of plant cell walls [47,48]. Characterization of the *V. dahliae* exoproteome revealed that at least 52 proteins participate in the pectin and cellulose degradation pathways [49]. Comparative genomics study has reported that *V. dahliae* encodes more CWDEs than other pathogenic fungi [48]. Several genes involved in cell wall degradation, such as *VdCUT11*, *VdPL3.1*, *VdPL3.3*, *VdXyn4*, *VdSSP1*, *VdSNF1,* and *VdFTF1*, have been found to be associated with the pathogenesis of *V. dahliae* [49,50,51,52,53,54]. In this study, we found that *VdPT1* has the ability to affect the expression of genes encoding CWDEs, which show significant down-regulation in the *VdPT1* deletion mutant. Due to the important role of CWDEs in the pathogenesis of *V. dahliae*, we concluded that the down-regulation of genes encoding CWDEs in the *VdPT1* deletion mutant is responsible for the decreased mycelial penetration ability and pathogenicity of *V. dahliae*.

### 3.3. Down-Regulation of Carbon, DNA, and Amino Acid Metabolism-Related Genes Is Responsible for the Slow Growth of V. dahliae

Stable carbon source metabolism is crucial for the occurrence of filamentous fungal diseases [55]. Fungi have a powerful carbohydrate degradation system, and the decomposition and metabolism of carbon sources provide not only energy sources for the life activities of these pathogens but also essential materials for their growth [56]. Glucose is the most preferred carbon [57], and its metabolism is closely related to glycolysis, the pentose phosphate pathway, and the glyoxylate cycle. In this study, we found down-regulation of multiple genes involved in glycolysis, the pentose phosphate pathway, and the glyoxylate cycle in the *VdPT1* deletion mutants, attributing to the decrease of glucose content in the mutants. The pentose phosphate pathway provides 5-P ribose for nucleic acid synthesis. GO and KEGG analysis found that DNA replication and amino acid biosynthesis-related terms and pathways were enriched in the down-regulated DEGs in *ΔVdPT1-1*, in line with the down-regulation of multiple genes of the pentose phosphate pathway. Taken together, the decreased glucose content in *ΔVdPT1-1* caused down-regulation of the expression levels of genes related to carbon metabolism, DNA replication, and amino acid biosynthesis, consequently resulting in the reduction of the energy and building materials required for the growth and development of *ΔVdPT1-1*. These results provided molecular evidence for the decreased colony expansion, delayed conidial germination, delayed mycelial growth, reduced conidial production, and reduced germination rate of *ΔVdPT1-1*. This slower growth and development may ultimately lead to a decrease in the pathogenicity of *ΔVdPT1*.

### 3.4. Up-Regulation of Glycerophospholipid Metabolism-Related Genes Is Responsible for the Enhanced Stress Resistance of V. dahliae

Trehalose has multiple physiological roles. In addition to storing energy, its stable structure is beneficial for forming a protective film on the cell surface. Trehalose has been implicated in the cellular response to numerous environmental stresses, such as high temperature, cold, starvation, oxidation, hypertonicity, and desiccation [19]. Under stressful conditions, fungi rapidly accumulate trehalose. Many fungal species show increased trehalose levels when in a dehydrated state [58]. Trehalose is required for the acquisition of tolerance to various stresses in the filamentous fungus *Aspergillus nidulans* [23]. Heat shock caused by external stimuli increases the trehalose content in *Saccharomyces cerevisiae* cells [24]. Significant accumulation of free trehalose was observed in dormant *Mycobacterium smegmatis* cells under acidified medium, and the trehalose breaks down when fungal spores recover from dormancy [10]. Overexpression of trehalase causes insufficient concentration of trehalose under stress conditions, thereby providing insufficient protein protection and increasing the risk of mortality [59]. Therefore, the increased trehalose content in the *VdPT1* deletion mutants observed in this study may be one of the reasons for the increased stress resistance of *V. dahliae*.

Although research has shown that trehalose enhances fungal stress resistance, the underlying molecular mechanisms are still unclear. Trehalose is considered to be involved in regulating cell membrane homeostasis, which prevents membrane damage during phase transition in the rehydration process [60,61]. The cell membrane is an essential barrier that protects cells from environmental stress [62,63]. Phospholipids constitute the bulk of the membrane lipids, and both the head group and the acyl chain could regulate cell membrane properties [64,65,66]. In this study, we found that genes related to glycerophospholipid metabolism, such as those encoding choline kinase and choline/ethanolamine phosphotransferase, were significantly up-regulated in *VdPT1* deletion mutants. A mutant of yeast choline kinase, the enzyme catalyzing the first enzymatic step in the synthesis of PC, had a decreased PC content in the membrane [67]. Choline/ethanolamine phosphotransferase catalyzes the final step in the synthesis of PC through the CDP choline pathway. Knocking out the genes encoding choline phosphotransferase (CPT1 or EPT1) in yeast leads to a decreased rate of PC synthesis and cannot reconstruct lipid synthesis [68,69]. Glycerophospholipids, including PC, PS, and PE, have been found to be involved in maintaining membrane integrity in eukaryote cells [70,71,72]. In this study, we observed an increased content of glycerophospholipids in *ΔVdPT1* caused by the up-regulation of genes involved in glycerophospholipid biosynthesis, which is likely to be the reason for the enhanced resistance of *ΔVdPT1* to NaCl stress due to better membrane integrity of *ΔVdPT1*.

In conclusion, the neutral trehalase VdPT1 is required for the colony expansion, conidial production and germination, and pathogenicity of *V. dahliae*. Mutation in *VdPT1* decreased glucose content and increased trehalose content in *V. dahliae*. This decrease of glucose content is caused by down-regulation of the genes related to carbon metabolism, DNA replication, and amino acid biosynthesis pathways, leading to slow colony expansion, delayed mycelial growth, and decreased conidial production and germination rates. The increased trehalose content in *ΔVdPT1* made the mutant strain more resistant to stress, likely due to enhanced membrane integrity and altered cell membrane components caused by the up-regulation of genes related to glycerophospholipid metabolism. Additionally, the mutation in *VdPT1* caused the down-regulation of several genes related to cell wall degradation, compromising mycelial penetration and the pathogenicity of the mutant strain (Figure 14).

## 4. Materials and Methods

### 4.1. Pathogen Strains and Plant Growth Conditions

The strongly pathogenic Vd991 strain of *V. dahliae* and its mutants (see below) were cultured on potato dextrose agar (PDA) at 25 °C in the dark for 7 days. *Agrobacterium tumefaciens* strains GV3101 and AGL1 used in this study were cultured in LB liquid medium containing appropriate antibiotics. Sterilized cotton seeds from the upland variety Xinluzao 7 were sown in pots and cotton plants were grown in a greenhouse with a light cycle of 16 h light and 8 h dark at 22–25 °C.

### 4.2. DNA, RNA Extraction, and cDNA Synthesis

Plant total DNA was extracted using a Plant DNA Isolation Mini Kit (Vazyme, Nanjing, China) for the detection of fungal biomass in cotton tissues. The total RNA of cotton samples was extracted using the EASYspinPlus Plant RNA Extraction Kit (Aidlab, Beijing, China). The total RNA of *V. dahliae* was extracted using the Fungal RNA Kit (Omega Inc., Guangzhou, China). After removing genomic DNA using DNase I, the RNA integrity was determined by agarose gel electrophoresis, and the RNA concentration and purity (OD260/280 ratio) were analyzed using a NanoDrop^®^ 2000 spectrophotometer (Thermo Scientific, Wilmington, DE, USA). The first strand of cDNA was synthesized using M-MLV reverse transcriptase and stored in a −20 °C refrigerator for qRT-PCR.

### 4.3. Bioinformatics Analysis

The nucleotide sequence of VDAG_03038 (*VdPT1*) was downloaded from the NCBI website (https://www.ncbi.nlm.nih.gov/ (accessed on 21 October 2022)). The structural domains of trehalase (PF01204, PF03633 and PF03632) were downloaded from the Pfam protein family database (https://www.ebi.ac.uk/interpro/entry/pfam/#table/ (accessed on 21 April 2023)) and used in finding trehalase-encoding genes in *V. dahliae* and other fungi using HMMER (v3.3.2) (https://www.ebi.ac.uk/Tools/hmmer/search/hmmsearch/ (accessed on 23 October 2022)). The candidate trehalase-encoding genes with an E-value < 1 × 10^−5^ were further confirmed using SMART (http://smart.Emblheidelberg.de/ (accessed on 24 October 2022)). The phylogenetic tree was constructed using the neighbor-joining method with MEGA 7.0 software and landscaped by the Interactive Tree Of Life online tool (https://itol.embl.de/ (accessed on 26 October 2022)). Multiple sequence comparisons were performed using Clustal Omega (https://www.ebi.ac.uk/Tools/msa/clustalo/ (accessed on 27 October 2022)) and embellished with Jalview software 2.11.3.0.

### 4.4. Generation and Confirmation of VdPT1 Deletion and Complementary Mutants

The pGKO_2_-Gate and pSULPH-mut-RG#PB vectors used for mutant generation were kindly donated by Dr Zhaosheng Kong, Institute of Microbiology, Chinese Academy of Sciences. To construct a deletion vector, a gene break structure targeting *VdPT1* was designed by replacing the target fragment (1146 bp) of *VdPT1* with a fragment from the hygromycin resistance gene (*HPH*) (1908 bp). The upstream (1066 bp) and downstream (1050 bp) flanking fragments of the target fragment were amplified using Vd991 genomic DNA as a template with paired primers *VdPT1*-Flank-5F/*VdPT1*-Flank-5R and *VdPT1*-Flank-3F/*VdPT1*-Flank-3R, respectively. The *HPH* fragment (1908 bp) was amplified from pUC-*HPH* plasmid DNA with *VdPT1*-*HPH*-F/*VdPT1*-*HPH*-R primers (Appendix A). All PCR products were generated using Phusion High-Fidelity DNA polymerase (NEB). The three fragments (1066, 1908, and 1050 bp) were then integrated into the pGKO_2_-Gate vector using the ClonExpress II one step Cloning Kit (Vazyme Biotech Co. Ltd., Nanjing, China) following the manufacturer’s instructions. The resulting vectors were transformed into Vd991 via a PEG-mediated transformation method [73]. The correct transformants were selected on PDA containing appropriate antibiotics and verified by PCR with primers Test-*VdPT1*-F1/R1, Test-*VdPT1*-F2/R2, and Test-*VdPT1*-F3/R3 (Appendix A).

To construct the complementary vector, a fragment about 4.7 kb in size containing sequences of the promoter (1.8 kb), coding region (2.2 kb), and terminator (0.7 kb) was amplified using Vd991 genomic DNA as the template with Promotor-F/*VdPT1*-R primers (Appendix A). The resulting PCR product was fused with the pSULPH-mut-RG#PB vector using the ClonExpress II one step Cloning Kit. The resulting vector was then transformed into the *VdPT1* deletion mutant (*ΔVdPT1*) via a PEG-mediated transformation method as described above and the correct transformants were verified by PCR with primers Test-*Hyg*-F/R (Appendix A).

### 4.5. Fungal Growth Assays

The wild-type Vd991 and its mutants were cultured in CM liquid medium at 25 °C for 5–7 days. Then the conidia of these strains were collected and adjusted to 1 × 10^3^ CFU/mL, 1 × 10^5^ CFU/mL, or 1 × 10^7^ CFU/mL with sterile water. For the growth assay, 10 µL of the prepared conidial suspension (1 × 10^7^ CFU/mL) was inoculated in the center of PDA, CM, and BMM solid media and incubated at 25 °C in the dark. PDA medium is a basic medium, which includes comprehensive and balanced nutrition that effectively meets the nutritional needs for microbial growth and development. CM culture medium contains various vitamins and trace elements that are beneficial for fungal growth. BMM medium is an improved medium for *V. dahliae* to produce a large number of microsclerotia [74]. Colony diameters were measured at 3-day intervals, and colony morphology was observed and photographed after 21 days of incubation.

For monoconidial growth observation, 5 µL of conidial suspension (1 × 10^3^ CFU/mL) was inoculated uniformly in the center of the PDA medium and observed under a stereomicroscope (Zeiss, Germany) after 12, 24 and 36 h of incubation. For mycelial growth observation, 5 µL of conidial suspension (1 × 10^5^ CFU/mL) was inoculated in the center of the PDA medium and observed under a light microscope (10 × 40 times) after 12, 24 and 48 h of incubation. A conidial germination rate assay was performed by inoculating 1 µL conidial suspension (1 × 10^5^ CFU/mL) onto PDA medium and incubating for 12 h at 25 °C in the dark. To evaluate the conidial production of different strains, 1 mL of conidial suspension (1 × 10^7^ CFU/mL) was inoculated into 100 mL of CM liquid medium and cultured at 25 °C for 3 days in the dark, and conidial production was determined using a hemocytometer plate under a light microscope.

For mycelial penetration assays, 10 µL of conidial suspension (1 × 10^7^ CFU/mL) was incubated on cellophane (φ = 0.45 μm) and placed on top of PDA medium for 7 days. Then the cellophane was removed, and the colony size on the PDA medium was investigated after 7 days of incubation. Three replicates were set up for each strain.

### 4.6. Stress Treatments

To test the sensitivity of different strains to abiotic stresses, 10 µL of conidial suspension (1 × 10^7^ CFU/mL) was incubated on PDA solid medium containing 0.7 M NaCl, 1.0 M KCl, 1.2 M sorbitol, or 200 mg/mL validamycin A (Val. A). In addition, 10 µL of conidial suspension (1 × 10^7^ CFU/mL) was incubated on PDA solid medium at a high temperature (HT, 30 °C) or a low temperature (LT, 15 °C). Colony diameters were measured at 3-day intervals, and colony morphology was observed and photographed after 15 days of incubation. Each experiment was repeated three times.

### 4.7. Pathogenicity Assays

When the first true leaves of cotton seedlings were fully expanded, cotton seedlings were inoculated with conidial suspension (1 × 10^7^ CFU/mL) of different strains via the root irrigation method [75]. Disease symptoms were recorded at 18 and 30 dpi (days post inoculation). The disease index (DI) was calculated based on a five-grade (0, 1, 2, 3 and 4) categorization of the disease symptoms of cotton seedlings (national standard number: GB/T28084-2011). The formula used for DI calculation was: DI = [(Σ disease grade × number of infected plants)/(total number of sampled plants × 4)] × 100%. Three replicates were conducted for each strain.

### 4.8. TRV (Tobacco Rattle Virus) Treatment

The interference fragment (343 bp) of *VdPT1* was amplified from Vd991 cDNA with gene-specific primers *VdPT1*-F/R (Appendix A), and then ligated into T-Vector pMD19 (simple). The interference fragment was cut from T-Vector by double digestion (*EcoR*1 and *Kpn*1) and then cloned into the pTRV2 vector with T_4_ DNA Ligase to generate the HIGS (host-induced gene silencing) vector pTRV2-*VdPT1,* which was then transformed into *Agrobacterium tumefaciens* strain GV3101 by electroporation. When the cotyledons of cotton seedlings were fully expanded, the cotton leaves were injected with TRV as previously described [75]. At about 10 days after injection, the newly emerged leaves of pTRV2-*GhCHLI*-treated cotton seedlings appeared to be bleaching. When the first true leaves of cotton seedlings were fully expanded, the cotton seedlings treated with pTRV2-*VdPT1* were infected with conidial suspension (1 × 10^7^ CFU/mL) via the root irrigation method. The DI was investigated at 14 dpi and 21 dpi.

### 4.9. Determination of Trehalase Activity, Trehalose and Glucose Content

For each strain, 1 mL of conidial suspension (1 × 10^7^ CFU/mL) was incubated in 100 mL CM liquid medium for 4 days and then the organism was collected by centrifugation to determine the trehalase activity, trehalose, and glucose content using commercial reagents provided by the Trehalase Activity Assay Kit (Solarbio, Beijing, China), Trehalose Assay Kit (Solarbio, Beijing, China), and Glucose Assay Kit (Solarbio, Beijing, China), respectively, following the manufacturer’s instructions for each kit.

### 4.10. RNA-Sequencing (RNA-Seq)

The wild-type Vd991 and *ΔVdPT1-1* deletion mutant strains were cultured in 200 mL CM liquid medium at 25 °C for 4 days and collected for RNA isolation for RNA-seq. Total RNA was isolated using the RNA simple total RNA Kit (Tiagen, Beijing, China) according to the manufacturer’s protocol. The RNA concentration, purity, and integrity were determined using a NanoDrop 2000 spectrophotometer (Thermo Scientific, Waltham, MA, USA). Frozen RNA samples were sent to Biomarker Technologies Co, LTD for RNA-seq. HISAT2 software 2.2.1 was used to compare the obtained clean reads to the reference genome database of *V. dahliae* (http://ftp.ebi.ac.uk/ensemblgenomes/pub/release-53/fungi/fasta/verticillium_dahliae/dna/ (accessed on 14 April 2022)). Differentially expressed genes were screened using DEGseq2 with the criteria of fold change > 2.0 and FDR < 0.01 [76]. R/topGO (2.18.0) used for GO enrichment analysis, with *p* value < 0.05 as the threshold. KEGG enrichment analysis of differentially expressed genes was performed using R/clusterProfiler (3.10.1) with a significance level of *p* value < 0.05.

### 4.11. Gene Expression Analysis

Roots, stems, and leaves of infected seedlings were sampled and used for RNA and DNA extraction. The primer pair *VdPT1*-qF1/R1 (Appendix A) was used to analyze the transcript level of *VdPT1* in infected seedlings by qRT-PCR and the cotton *tubulin* gene was used as an internal. The *V. dahliae*-specific primers ITS1-F and ST-Ve1-R were used for fungal biomass measurement and the cotton *GhUBQ7* gene (DQ116441.1) was used as an internal standard (Appendix A) [75]. qRT-PCR assays were performed using SYBR Green mix with an initial program of 95 °C, 10 s and a cycling program of 60 °C, 15 s; 72 °C, 20 s; for 40 cycles. qRT-PCR reactions were performed on a Roche LightCycler 480 II instrument and the results were analyzed using the 2^−ΔΔCT^ method [77]. Ten genes were selected for qRT-PCR to test the reliability of the transcriptome data, and the primers used in qRT-PCR are shown in Appendix A.

### 4.12. Cell Membrane Integrity Analysis

To assess the cell membrane integrity of different *V. dahliae* strains, conidial suspension (1 × 10^7^ CFU/mL) was inoculated into 100 mL of CM liquid medium containing 0 M or 0.7 M NaCl and incubated for 4 days at 25 °C with 220 rpm/min shaking. Samples were centrifuged, washed twice with PBS, and adjusted to 1 × 10^7^ CFU/mL. Then the diluted sample (200 μL) was incubated with 4 μL of propidium iodide (PI) (Solarbio, Beijing, China) at 37 °C in the dark. After 30 min, 600 μL of physiological saline was added and mixed by blowing to terminate staining. A flow cytometer (Luminex, Model: Guava EasyCyte) using the Red-B channel was used to detect apoptotic changes and data were analyzed with FlowJo.

### 4.13. Lipidomics Analysis

The samples used for lipidomics analysis were the same as those for RNA-seq. The collected fungal samples were mixed with 200 μL water and 20 μL internal lipid standard mixture, then supplemented with 800 μL methyl tert-butyl ether (MTBE). After mixing, 240 μL pre-cooled methanol was added and mixed thoroughly. The suspension was subjected to ultrasound in a low-temperature (4 °C) water bath for 20 min. After 30 min at room temperature, the sample was centrifuged at 14,000× *g* for 15 min at 10 °C. The upper layer of the organic phase was blow-dried with nitrogen gas. The dried samples were dissolved in 200 μL 90% isopropanol/acetonitrile solution and used in the following analysis.

Analysis of lipidomics was performed using liquid chromatography-mass spectrometry (LC-MS; Thermo Scientific™). Samples were separated using a UHPLC Nexera LC-30A ultra-high-performance liquid chromatography system with a C18 column with gradient elution at 45 °C and a rate of infusion of 300 μL/min. The mobile-phase gradient was formed by buffer A ((acetonitrile:water = 6:4, *v*/*v*) + 0.1% formic acid + 0.1 mM ammonium formate) and buffer B ((acetonitrile:isopropanol = 1:9, *v*/*v*) + 0.1% formic acid + 0.1 mM ammonium formate). The gradient elution procedure was as follows: 0–3.5 min, buffer B maintained at 40%; 3.5–13 min, buffer B changed linearly from 40% to 75%; 13–19 min, buffer B changed linearly from 75% to 99%; 19–24 min, buffer B maintained at 40%. During the entire analysis process, the sample was placed in a 10 °C automatic sampler. The sample was separated by UHPLC and subjected to mass spectrometry analysis using the Q Executive series mass spectrometer (Thermo Scientific™). The positive and negative ion modes of electro-spray ionization (ESI) were used, and the spray voltage was set at 3.0 kV for detection. The mass-to-charge ratio of lipid molecules and lipid fragments were collected using the following method: 10 fragment maps (MS2 scan, HCD) were collected after each full scan. MS1 has a resolution of 70,000 at M/Z 200, and MS2 has a resolution of 17,500 at M/Z 200. LipidSearch was used for peak recognition, peak extraction, and identification (secondary identification) of lipid molecules.

## Figures and Tables

**Figure 1 ijms-25-00294-f001:**
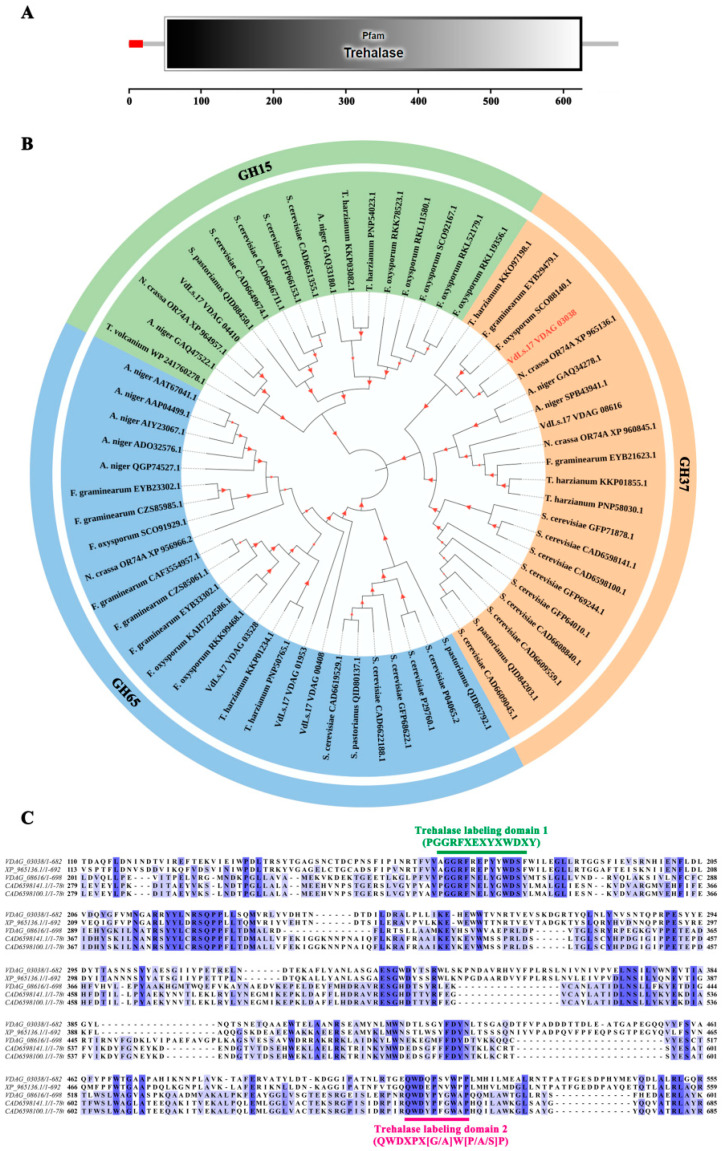
Functional structural domains of VdPT1, phylogenetic tree, and multiple sequence alignment of trehalases. (**A**) The trehalase domains in VdPT1 predicted based on Pfam. (**B**) The phylogenetic tree of trehalases from nine microbes constructed by MEGA 7.0 through the Neighbor-joining method (NJ) and the bootstrap test repeated 1000 times. The trehalases were grouped into three clades, which were distinguished by different colors. The nine species are as follows: *V. dahliae*—*Verticillium dahliae*; *T. volcanicus*—*Thermoplasma volcanicus* (bacteria); *A. niger*—*Aspergillus niger*; *N. crassa*—*Neurospora crassa*; *S. cerevisiae*—*Saccharomyces cerevisiae*; *S. pastorianus*—*Saccharomyces pastorianus*; *T. harzianum*—*Trichoderma harzianum*; *F. oxysporum*—*Fusarium oxysporum*; *F. graminearum*—*Fusarium graminearum*. (**C**) Alignment of four selected trehalases of the GH37 subfamily, including VdPT1 (VDAG_03038). The conserved trehalase labeling domain 1 (PGGRFXEXYXWDXY) and domain 2 (QWDXPX[G/A]W[P/A/S]P) are marked by green and magenta bars, respectively.

**Figure 2 ijms-25-00294-f002:**
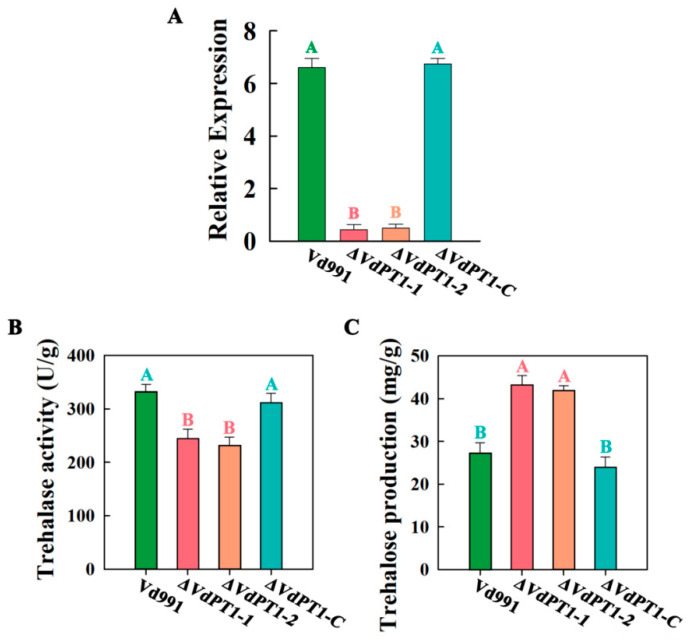
The expression levels of VdPT1, trehalase activity, and trehalose content in different *V. dahliae* strains. (**A**) The expression levels of *VdPT1* in different strains as determined by qRT-PCR. (**B**) Trehalase activity in different strains. (**C**) Trehalose content in different strains. Data were statistically analyzed using IBM SPSS Statistics 26.0. Significant differences between different treatments were analyzed using Duncan’s multiple-range tests (different letters above the error bars indicate statistically different at *p* < 0.01) for one-way ANOVA.

**Figure 3 ijms-25-00294-f003:**
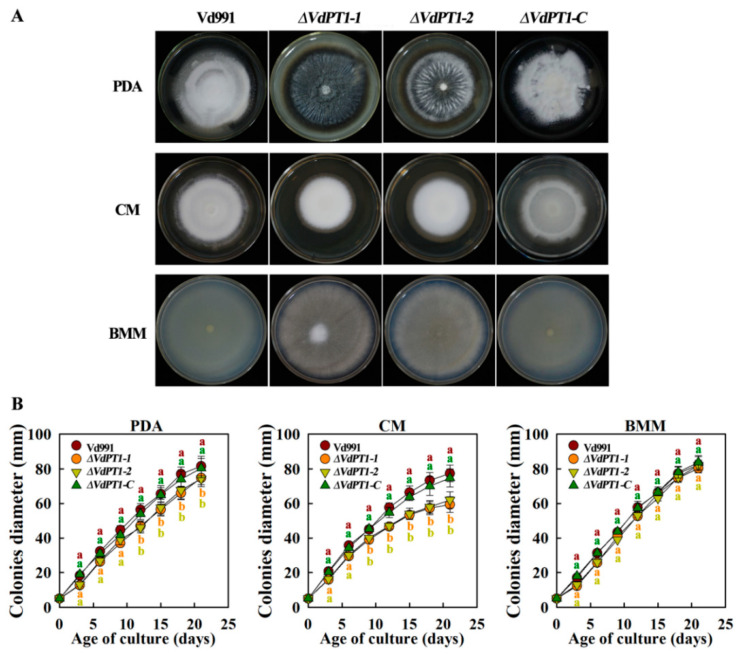
Colony morphology and growth rates of different *V. dahliae* strains. All strains were incubated on PDA, CM, or BMM medium. (**A**) Colony morphology of Vd991, deletion mutants (*ΔVdPT1*-*1*, *ΔVdPT1*-*2*), and a complementary strain (*ΔVdPT1*-*C*) after 21 days of incubation on media. (**B**) Colony growth rates of different strains on PDA, CM, or BMM medium. Values were means ± SD from three replicates. The data were statistically analyzed using IBM SPSS statistics 26.0. Significant differences between different treatments were analyzed using Duncan’s multiple-range tests. Letters above the error bars indicate statistically significant differences at *p* < 0.05 from one-way ANOVA tests.

**Figure 4 ijms-25-00294-f004:**
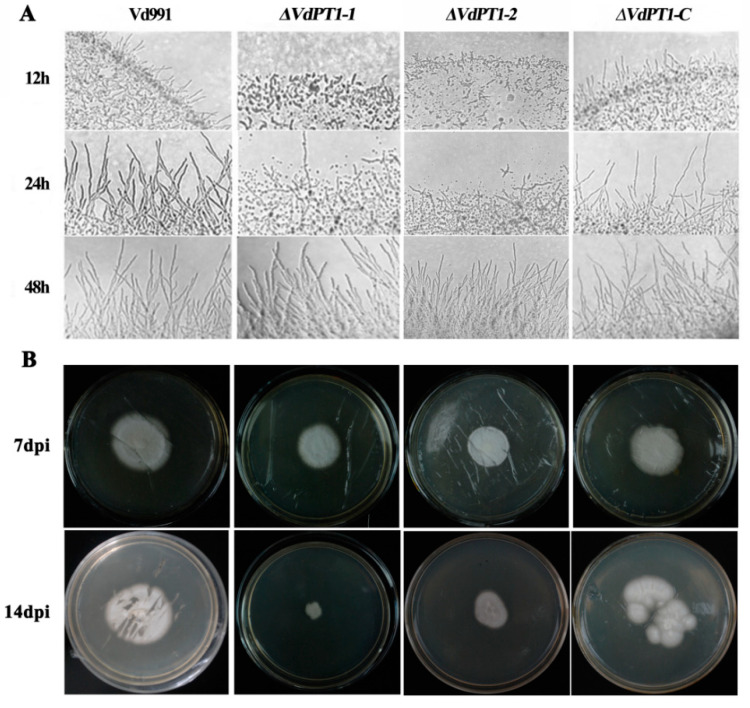
Detection of mycelial growth and penetration of different *V. dahliae* strains. (**A**) Mycelial growth of different strains grown on PDA medium after 12, 24 and 48 h of inoculation. (**B**) Cellophane penetration assay. The top panel shows different strains grown on PDA medium covered with cellophane at 7 days post inoculation (7 dpi), and the bottom panel shows the growth of different strains at 7 days after removal of cellophane (14 dpi).

**Figure 5 ijms-25-00294-f005:**
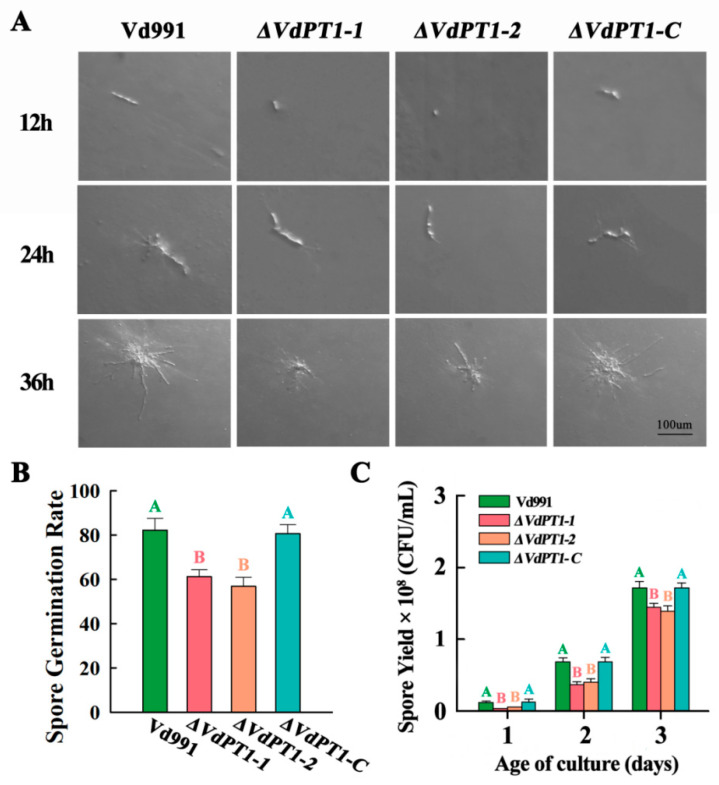
Comparison of monoconidial germination, conidial production, and germination rate of different *V. dahliae* strains. (**A**) Micrographs of monoconidial growth of different strains on PDA medium. (**B**) Conidial germination rates of different strains on PDA medium after 12 h of incubation. (**C**) Conidial production of different strains in CM liquid medium after 24, 48, and 72 h of incubation. The results were based on at least three independent experiments. Data were statistically analyzed using IBM SPSS statistics 26.0 and significance analysis was performed using Duncan’s multiple range tests. Different letters above the error bars indicate statistically significant differences at *p* < 0.01 from one-way ANOVA tests.

**Figure 6 ijms-25-00294-f006:**
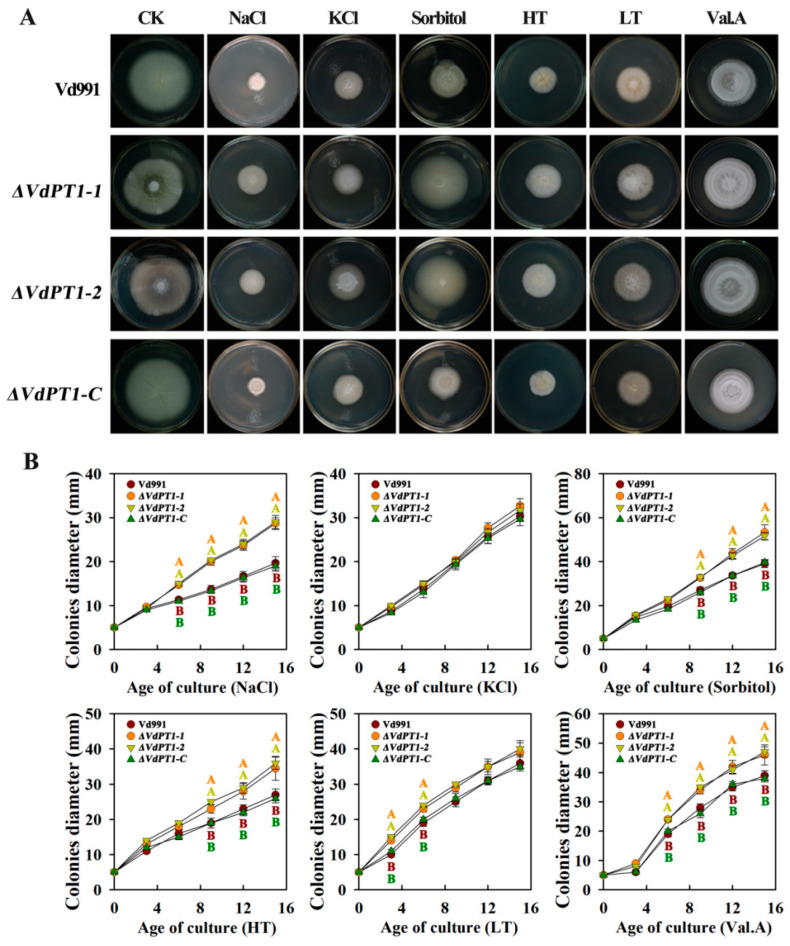
Comparison of colony morphology and growth rates of different *V. dahliae* strains under various stress conditions. (**A**) Colony morphology of different strains under various stress conditions. (**B**) Colony growth rates of different strains under various stress conditions. HT and LT indicate high and low temperature, respectively. Data were statistically analyzed using IBM SPSS Statistics 26.0. Significant differences between different treatments were analyzed using Duncan’s multiple range tests (different letters above the error bars indicate statistically significant differences at *p* < 0.01) from one-way ANOVA tests.

**Figure 7 ijms-25-00294-f007:**
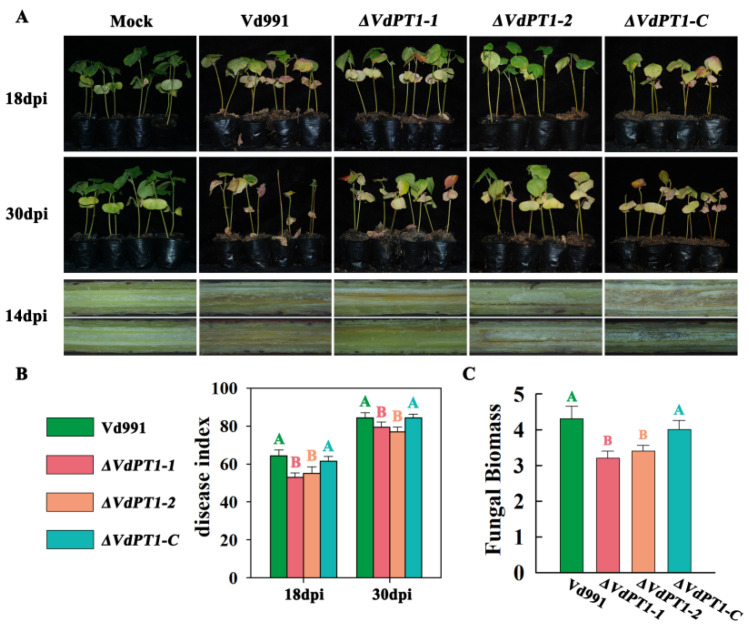
Comparison of pathogenicity of different *V. dahliae* strains. (**A**) Disease symptoms of cotton plants infected with different strains at 18 dpi and 30 dpi. Vascular browning observation was performed at 14 dpi. (**B**) Disease index of cotton plants infected with different strains at 18 dpi and 30 dpi. (**C**) qRT-PCR assay of fungal biomass in cotton plants treated with different strains at 21 dpi. Data were statistically analyzed using IBM SPSS statistics 26.0. Significant differences between different treatments were analyzed using Duncan’s multiple-range tests. Different letters above the error bars indicate statistically significant differences at *p* < 0.01 from one-way ANOVA tests.

**Figure 8 ijms-25-00294-f008:**
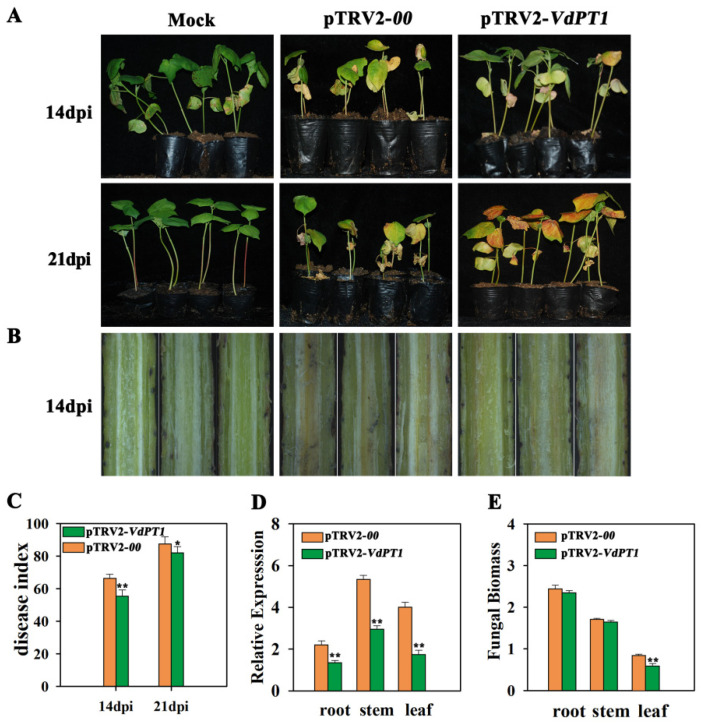
Functional assessment of *VdPT1* in the pathogenicity of *V. dahliae* by TRV-based HIGS. (**A**) Fungal infection symptoms of cotton seedlings treated with pTRV2-*00* or pTRV2-*VdPT1* at 14 and 21 dpi. (**B**) Comparison of vascular browning in the stem segments of HIGS cotton seedlings at 14 dpi. (**C**) Disease index of HIGS cotton seedlings at 14 and 21 dpi. (**D**) qRT-PCR assay of *VdPT1* expression in root, stem and leaf of HIGS cotton seedlings at 21 dpi. Cotton *tubulin* gene was used as the internal reference gene. (**E**) qRT-PCR assay of fungal biomass in HIGS cotton seedlings at 21 dpi. The data were statistically analyzed used IBM SPSS statistics 26.0. Statistical significance was determined using Student’s *t*-test. ** and * above the error line indicate a significant difference between HIGS cotton seedlings and pTRV2-*00* control at *p* < 0.01 and *p* < 0.05, respectively.

**Figure 9 ijms-25-00294-f009:**
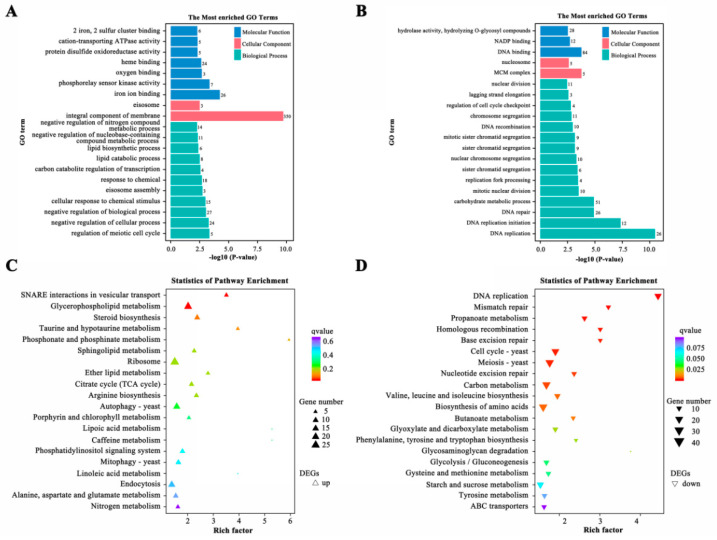
Enrichment analysis of the genes differentially expressed between wild-type (Vd991) and *ΔVdPT1* strains of *V. dahliae*. (**A**) GO enrichment analysis of up-regulated DEGs in the *ΔVdPT1*-*1* strain. (**B**) GO enrichment analysis of down-regulated DEGs in the *ΔVdPT1*-*1* strain. (**C**) KEGG pathway analysis of up-regulated DEGs in the *ΔVdPT1*-*1* strain. (**D**) KEGG pathway analysis of down-regulated DEGs in the *ΔVdPT1*-*1* strain.

**Figure 10 ijms-25-00294-f010:**
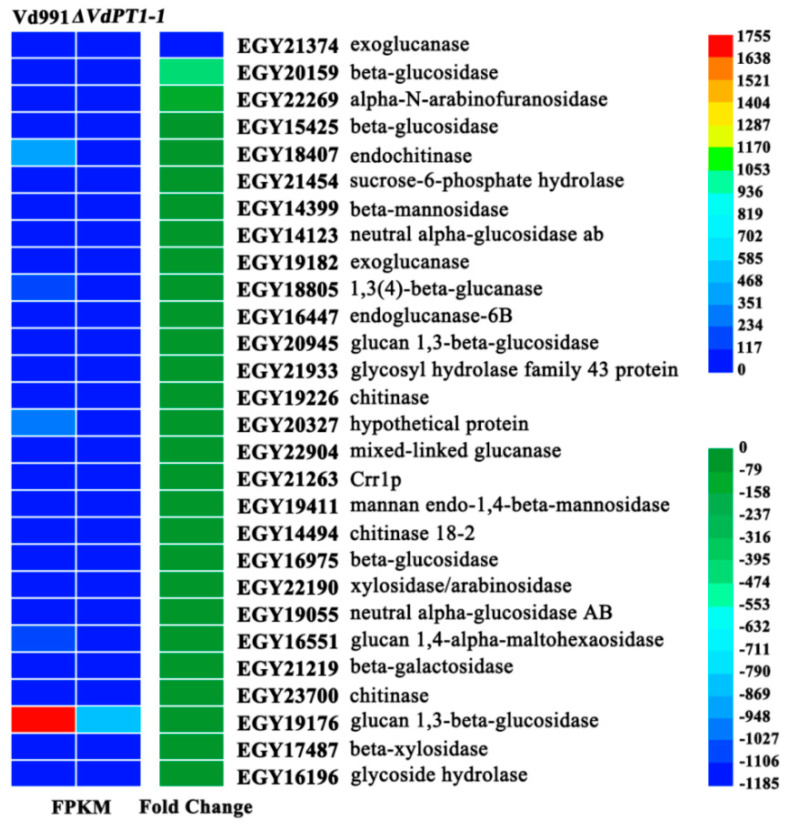
Heatmap showing the expression levels of DEGs related to hydrolase activity and the hydrolysis of O-glycosyl compounds.

**Figure 11 ijms-25-00294-f011:**
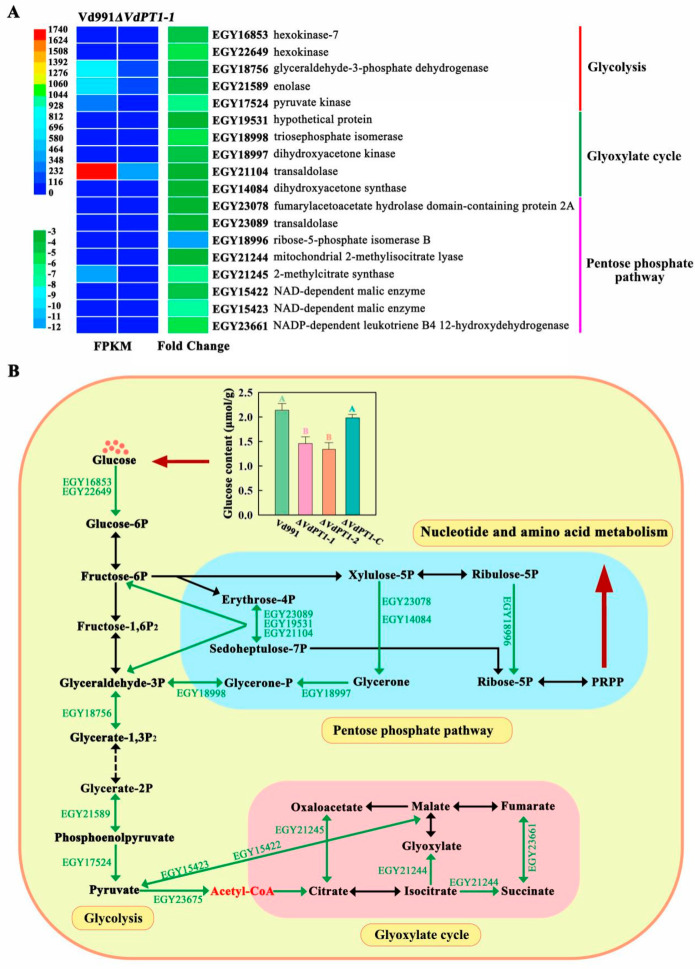
Heatmap showing the expression levels of DEGs related to carbon metabolism and related pathways.(**A**) Heatmap of DEGs related to glycolysis, the pentose phosphate pathway, and the glyoxalate cycle. (**B**) A diagram showing carbon metabolism-related pathways. The inset shows the glucose content in different *V. dahliae* strains. Data were statistically analyzed using IBM SPSS Statistics 26.0. Significant differences between different treatments were analyzed using Duncan’s multiple range tests. Different letters above the error bars indicate statistically significant differences at *p* < 0.01 from one-way ANOVA tests.

**Figure 12 ijms-25-00294-f012:**
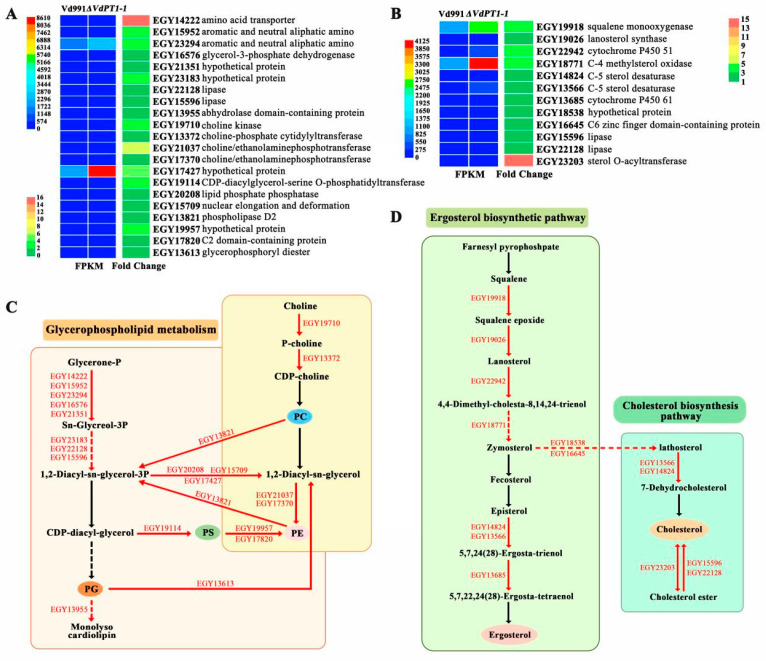
Heatmap and network diagram of DEGs related to glycerophospholipid metabolism and steroid biosynthesis. (**A**) Heatmap of DEGs related to glycerophospholipid metabolism. (**B**) Heatmap of DEGs related to steroid biosynthesis. (**C**) A diagram showing the network of DEGs related to glycerophospholipid metabolism. (**D**) A diagram showing the network of DEGs related to steroid biosynthesis.

**Figure 13 ijms-25-00294-f013:**
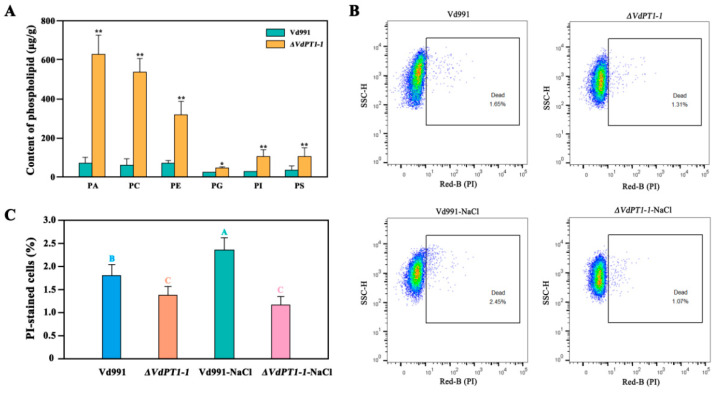
Comparison of glycerophospholipid content and cell membrane integrity between wild-type (Vd991) and deletion mutant *ΔVdPT1-1*. (**A**) The content of glycerophospholipids in wild-type and the *ΔVdPT1-1* mutant. (**B**) Membrane integrity of wild-type and the *ΔVdPT1-1* mutant at 0 M NaCl and 0.7 M NaCl examined by flow cytometry analysis. The boxed area is the percentage of PI-stained cells or dead cells caused by impaired cell membranes. (**C**) Quantification of membrane integrity in the wild-type and the *ΔVdPT1-1* mutant at 0 M NaCl and 0.7 M NaCl. All data are the mean values of three independent experiments. Data were statistically analyzed using IBM SPSS statistics 26.0. Statistical significance was determined using Student’s *t*-test. Asterisks (*/**) above the error bars indicated significant difference at *p* < 0.05 or *p* < 0.01 between wild-type (Vd991) and deletion mutant (*ΔVdPT1-1*). Significant differences in different treatments were analyzed using Duncan’s multiple-range tests. Different letters above the error bars indicate statistically significant differences at *p* < 0.01 from one-way ANOVA tests.

**Figure 14 ijms-25-00294-f014:**
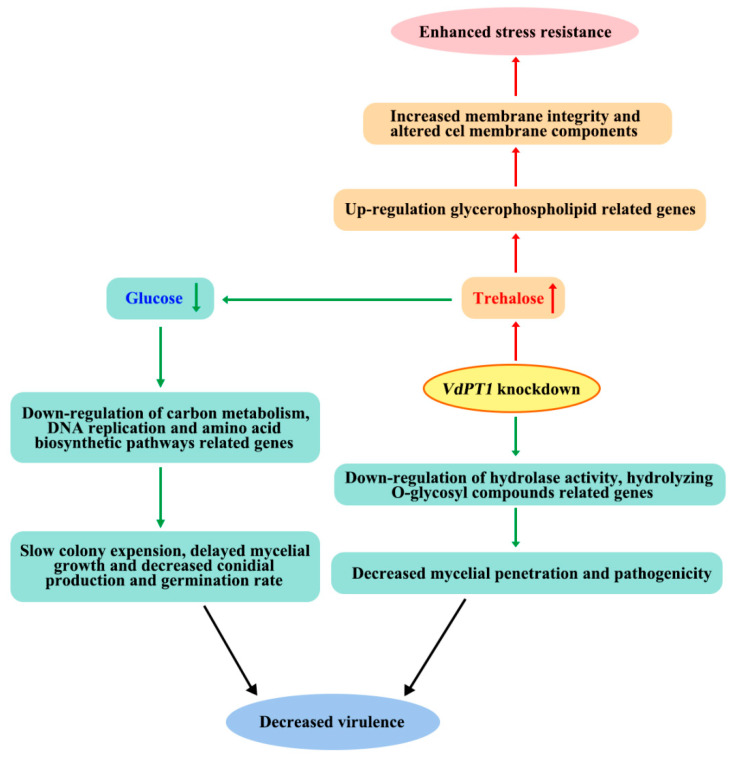
A working model for the role of *VdPT1* in the growth and development, pathogenicity, and stress resistance of *V. dahliae*.

## Data Availability

Raw sequencing data can be accessed through the Gene Expression Omnibus with the accession number PRJNA1007260.

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
