# Peer review of "VdPT1 Encoding a Neutral Trehalase of Verticillium dahliae Is Required for Growth and Virulence of the Pathogen"

_ijms, 2023, doi:10.3390/ijms25010294_

Round 1

Reviewer 1 Report

Comments and Suggestions for Authors

The authors identify a V. dahliae trehalase encoding-gene (VdPT1) that is up-regulated after sensing cotton root exudates from susceptible cotton variety.  They create VdPT1 deletion mutants (ΔVdPT1) and assess their growth and infectivity characteristics to determine if the VdPT1 gene product functions as a virulence factor.  The ΔVdPT1 strains showed slow growth and reduced infectivity, but had increased resistance to stress (trehalose is often used under temp/osmotic stresses).  They also showed that host induced VdPT1 silencing reduced fungal biomass and enhanced cotton resistance against.  They also note that in the ΔVdPT1 strain there were several down-regulated genes involved in carbon metabolism, DNA replication and amino acid biosynthesis.  The authors use proper methods, the paper is well-done, easily interpretable, the data were objectively and correctly analyzed, and overall the data justify the conclusions, but there are some points to consider. 

11)      Was the VdPT1 gene upregulated in root exudates from resistant cotton varieties? 

22)      The infection/virulence effects seem rather slight when comparing the DVdPT1-1/VdPT1-2 strains to the controls, and grow of the DVdPT1-1/VdPT1-2 strains is clearly not as good as wild type, so can the authors more carefully explain/highlight the difference between how to differentiate/separate a virulence factor from the loss of a normal metabolic function (e.g. lower glucose level) that may prevent/slow growth?   

33)      The authors should explain the significance of each media type used in Figure 3 and throughout the manuscript in either the methods or results sections

44)      Are other trehalase genes upregulated in the DVdPT1-1/VdPT1-2 strains? 

Comments on the Quality of English Language

minor editing 

Author Response

Dec. 12, 2023 

To

Editor

Dear Editor, 

We are grateful to you for your kind letter of encouragement, along with the constructive comments of the reviewers concerning our manuscript titled "VdPT1 encoding a neutral trehalase of Verticillium dahliae is required for growth and virulence of the pathogen" (Manuscript ID: 2737356). We have considered all the comments of the reviewers and editors and substantially revised our manuscript. We hope, with these modifications and improvements, the quality of our manuscript would meet the publication standard of International Journal of Molecular Sciences.

Our point-by-point responses to the two reviewers’ comments have been enclosed below, and all the changes to our ms have been highlighted by colored text in the revised version for your viewing. 

Once again, many thanks for your constructive comments, which are highly helpful in improving the quality of our manuscript.

We are looking forward to hearing from you.

Sincerely,

Jie Sun

(Corresponding Author)

Reviewer’s comments and our responses

  • Was the VdPT1 gene upregulated in root exudates from resistant cotton varieties?

Our response: Many thanks to the reviewer for this question. The VdPT1 gene (VDAG_03038) was obviously upregulated in root exudates from susceptible cotton variety, but not in root exudates from resistant cotton varieties. Readers can find this gene in Table 6 of our previously published paper (Zhang et al., 2020), which includes all genes that are only upregulated in the root exudates from susceptible variety. We have described this content in the our manuscript, please refer to Line 77-79.

  • The infection/virulence effects seem rather slight when comparing theDVdPT1-1/VdPT1-2 strains to the controls, and grow of theDVdPT1-1/VdPT1-2 strains is clearly not as good as wild type, so can the authors more carefully explain/highlight the difference between how to differentiate/separate a virulence factor from the loss of a normal metabolic function (e.g. lower glucose level) that may prevent/slow growth?

Our response: Many thanks to the reviewer for this question. Compared with the wild type, the mutant (ΔVdPT1-1/VdPT1-2) showed slower growth and reduced pathogenicity, which are all phenotypes of the mutant. In fact, the slower growth of mutants may ultimately lead to a decrease in pathogenicity of ΔVdPT1. We have added the sentence into Discussion 3.3 section, please refer to Line 445-446 in revised manuscript. The reader also can see their difference from Figure 14 (A working model for the role of VdPT1 in the growth and development, pathogenicity and stress resistance of V. dahliae).

These two phenotypes (slower growth and reduced pathogenicity) are different, and in our discussion, we analyzed the reasons for the slow growth of mutants and the weakened pathogenicity, respectively.

For example, Line 439-445, ‘the decreased glucose content in ΔVdPT1-1 caused down-regulation of the expression levels of genes related to carbon metabolism, DNA replication and amino acid biosynthesis, consequently resulting in the reduction of the energy and building materials required for the growth and development of ΔVdPT1-1. The results provided molecular evidence for the decreased colony expansion, delayed conidial germination and mycelial growth, reduced conidial germinate rate and production of ΔVdPT1-1.’

Line 422-423, ‘Due to the important role of CWDEs in the pathogenesis of V. dahliae, we concluded that the down-regulation of the genes encoding CWDEs in the VdPT1 deletion mutant is responsible for the decreased mycelial penetration ability and pathogenicity of V. dahliae.’ 

  • The authors should explain the significance of each media type used in Figure 3 and throughout the manuscript in either the methods or results sections.

Our response: Many thanks to the reviewer for this comment. Our purpose of using three different culture media is relatively simple, just to compare the growth rate of mutant and wild-type strains. PDA medium is a basic medium, which includes comprehensive and balanced nutrition that effectively meets the nutritional needs of microbial growth and development. CM culture medium contains various vitamins and trace elements that are beneficial for fungal growth. BMM medium is an improved medium for V. dahliae to produce a large number of microsclerotia (Neumann MJ and Dobinson KF, 2003). We have added these contents into methods 4.5 section. Please refer to Line 558-562.

  • Are other trehalase genes upregulated in theVdPT1-1/VdPT1-2 strains?

Our response: Many thanks to the reviewer for this question. We apologize for this oversight as we did not detect the expression of other trehalase genes. But we detected changes in the activity of trehalase, which is beneficial for describing and explaining our results. Please refer to Figure 2.

Reviewer 2 Report

Comments and Suggestions for Authors

The article is well written but needs some minor corrections as shown on the manuscript.

I did not check for plagiarism.

Comments on the Quality of English Language

Minor corrections on English language is required.

Author Response

Dec. 12, 2023 

To
reviewer
International Journal of Molecular Sciences
Dear reviewer, 
We are grateful to you for your kind letter of encouragement, along with the constructive comments of the reviewers concerning our manuscript titled "VdPT1 encoding a neutral trehalase of Verticillium dahliae is required for growth and virulence of the pathogen" (Manuscript ID: 2737356). We have considered all the comments of the reviewers and editors and substantially revised our manuscript. We hope, with these modifications and improvements, the quality of our manuscript would meet the publication standard of International Journal of Molecular Sciences.
Our point-by-point responses to the two reviewers’ comments have been enclosed below, and all the changes to our ms have been highlighted by colored text in the revised version for your viewing. 
Once again, many thanks for your constructive comments, which are highly helpful in improving the quality of our manuscript. 
We are looking forward to hearing from you.

Sincerely,
Jie Sun
(Corresponding Author)
